# How Epstein Barr Virus Causes Lymphomas

**DOI:** 10.3390/v16111744

**Published:** 2024-11-06

**Authors:** Ya-Fang Chiu, Khongpon Ponlachantra, Bill Sugden

**Affiliations:** 1Department of Microbiology and Immunology, Chang Gung University, Taoyuan 33302, Taiwan; yfchiu@mail.cgu.edu.tw; 2Graduate Institute of Biomedical Sciences, Chang Gung University, Taoyuan 33302, Taiwan; 3Research Center for Emerging Viral Infections, Chang Gung University, Taoyuan 33302, Taiwan; 4Division of Infectious Diseases, Department of Medicine, New Taipei Municipal Tucheng Hospital, New Taipei City 236017, Taiwan; 5Department of Laboratory Medicine, Chang Gung Memorial Hospital, Linkou 33305, Taiwan; 6School of Biomolecular Science and Engineering, Vidyasirimedhi Institute of Science and Technology, Rayong 21210, Thailand; khongpon.p@gmail.com; 7McArdle Laboratory for Cancer Research, University of Wisconsin-Madison, Madison, WI 53705, USA

**Keywords:** EBV, lymphomas

## Abstract

Since Epstein–Barr Virus (EBV) was isolated 60 years ago, it has been studied clinically, epidemiologically, immunologically, and molecularly in the ensuing years. These combined studies allow a broad mechanistic understanding of how this ubiquitous human pathogen which infects more than 90% of adults can rarely cause multiple types of lymphomas. We survey these findings to provide a coherent description of its oncogenesis.

## 1. Introduction

Epstein–Barr Virus (EBV) causes lymphomas, “cancers that arise from the clonal proliferation of B cells, T cells and natural killer (NK) cells” [1]. By EBV causing lymphomas, we mean that the lymphomas that have EBV genomes in the tumor cells are dependent on EBV for their development. We know EBV-positive lymphomas depend on the virus because its genome is present as a circular replicon and is lost from cells if it does not provide them with one or more advantages. The replication and partitioning of EBV plasmids in live cells have been visualized [2] and found to be imperfect. Each cell division, each circular genome, has an 84% chance of being synthesized and an 88% chance of being partitioned faithfully (*ibid.*). With each generation, some of the EBV is therefore lost from the division of cells. Only if the cells that retain it outgrow those that have lost it will it be found in growing cells. EBV-positive lymphoma biopsies generally have circular EBV genomes in most/all cells, indicating that the virus provides these tumors benefits in survival/growth or both [3]. Occasionally, EBV DNA becomes integrated into the host genome of a lymphoma cell. The sequencing of the EBV’s genome in 128 samples of NK/T cell lymphomas in France and Japan, for example, identified that 6% had deletions of *OriP*, the cis-acting viral element required for EBV DNA synthesis [4]. These viral genomes could only be maintained in the proliferating tumor cells if they were integrated.

At least five types of lymphomas are associated causally with EBV: Burkitt lymphoma, NK/T cell lymphoma, Hodgkin’s lymphoma, Diffuse Large B Cell Lymphoma (DLBCL), and Primary Effusion Lymphoma (PEL). While each of these lymphomas have cases that are EBV-positive, the fraction of cases with EBV in the tumor cells varies. For example, more than 90% of Burkitt lymphomas are EBV-positive in Central Africa, where it is endemic, but only 29% are positive in the US [5,6]. About 90 to 100% of nasal NK/T cell lymphomas are EBV-positive throughout the world [7]. About 40% of Hodgkin’s, 5 to 10% of DLBCL, and 80–90% of PELs are EBV-positive [8]. These different frequencies of being virus-positive for lymphomas associated with EBV have two implications. First, the clinical diagnoses do not distinguish between EBV-positive and virus-negative lymphomas of the same type. Second, given that we know EBV contributes causally to the virus-positive lymphomas, the EBV-negative variants must have some genetic changes to provide the oncogenic functions that EBV uses to drive the virus-positive lymphomas.

There are now about 8.2 billion people in the world and EBV causes about 50,000–120,000 new cases of lymphomas per year in the world [9], meaning these cancers are rare. This rarity is particularly striking because it is estimated that greater than 90% of all of us are infected with EBV, indicating that only one per hundred thousand people develop a lymphoma caused by EBV per year. This rarity leads to the question of what, in addition to infection with EBV, leads to the development of its associated lymphomas. There are at least three possible answers to the question that merit consideration: 1. Rare variants of EBV are sufficient to cause lymphomas; 2. Genetic variants in people coupled to infection with EBV contribute to lymphomagenesis; and 3. Co-factors in concert with EBV infection foster lymphomagenesis.

### 1.1. Rare Variants of EBV Are Not Sufficient to Cause Lymphomas

If a rare variant of EBV is sufficient to cause a lymphoma, then that lymphoma would be expected to occur in a cluster in space and time as that variant infects people locally. While such clustering has been reported, its interpretation is complex. EBV-associated lymphomas are rare and viral variants cluster geographically. In a recent study, researchers extended the sequencing of EBV genomes in 54 patients with NK/T-cell lymphoma in Guatemala to compare it to data from a similar 101 patients in Southeast Asia. They concluded “that the previously suggested variants were not disease-specific but rather common to EBV in East Asia” [10]. No single variant of EBV has been found to be sufficient to cause a lymphoma in otherwise healthy people.

### 1.2. Genetic Variants Coupled with EBV Contribute to Lymphomagenesis

Human genetic variants coupled with infection by EBV do contribute to lymphomagenesis. Two of these mutations, while rare, have been studied in detail. Duncan’s syndrome is an X-linked immunodeficiency with mutations in either the *SH2D1A* gene or *XIAP* gene [11,12]. Mutations in the *SH2D1A* gene that encode the SLAM-associated protein (SAP) lead to immune failures in recognizing and controlling EBV-infected cells. These failures have been uncovered in studies of female carriers of these mutations who are healthy while males succumb to EBV-induced diseases including lymphomas. Only the SAP-positive CD8^+^ T cells from female carriers respond to EBV peptides; SAP-negative CD8^+^ T cells cannot [13]. This failure reflects the absence of inhibitory signaling by SAP on the CD8^+^ T cells following the recognition of antigens of B cells and explains *SH2D1A* being essential to control infections by EBV (*ibid.*).

Mutations in the Magnesium transporter 1, *MAGT1*, also limit the T cell control of EBV-infected B cells leading to the X-linked disease, XMEN, predisposing patients to EBV-associated lymphomas [14,15]. Studies of cells from XMEN patients have shown that both T cells and NK cells have low intracellular concentrations of free Mg^2+^ and decreased capacities to kill autologous EBV-positive B cells [16]. The supplementation of culture media with Mg^2+^ for T cells and NK cells from patients led to the increased expression of the NKG2D receptor on both cell types and increased the cytotoxicity of T cells (*ibid.*). The diminished expression of NKG2D on NK cells in XMEN patients is likely to contribute to EBV lymphomagenesis, too. Both T cells and NK cells control EBV infections [17]. The NK cell killing of EBV-infected cells from healthy donors is directly regulated by EBV. Its EBNA1 protein binds to the promoter of ULBP1, a ligand for NKG2D, and decreases it expression and decreases the NK cell killing of EBV-infected cells [18].

### 1.3. Co-Infections Augment Lymphomagenesis

Co-infections with multiple different pathogens can promote EBV’s lymphomagenesis by modifying the viral-host environment to limit some immune responses [19]. These co-infecting pathogens include both protozoa and viruses. A host’s immune response to EBV is essential to control it. For example, during primary infections, children in The Gambia have been shown to develop 6–16% of CD8^+^ T cells responding to an EBV-specific epitope [20]. Co-infections that limit a person’s ability to mount an effective immune response to EBV are therefore likely important in supporting its lymphomagenesis.

#### 1.3.1. Malaria Co-Infection

Malaria is one of the most prevalent infectious diseases worldwide with approximately 250 million cases in 2022 according to the World Health Organization. It is caused by parasites from the genus *Plasmodium* and transmitted from humans to humans through the bites of infected Anopheles mosquitoes [21].

Studies of children with Burkitt lymphoma in regions of Africa where it is endemic have found that the number of malaria parasites, nutritional status, and EBV viral load correlate with the incidence of the lymphoma [22]. The number of EBV-infected B cells in young patients in The Gambia also correlates with their malarial status, being five times higher during acute malarial infection [23]. Malarial infection where Burkitt lymphoma is endemic appears not to limit a host’s immune response generally but does decrease its ability to control infection with EBV [24]. For example, children with endemic Burkitt lymphoma have decreased IFN-γ-producing T cell responses to EBNA1 peptides compared to controls but higher levels of these T cell responses to a malarial antigen [25]. These children also have different sets of NK cells as assessed by the distributions of subsets of killer immunoglobulin-like receptor genes on these NK cells [26]. These differences are likely to underlie a decreased NK cell response that contributes to EBV’s pathogenesis.

#### 1.3.2. Human Immunodeficiency Virus (HIV) Co-Infection

Individuals living with acquired immunodeficiency syndrome (AIDS) were found to have a 10–100-fold increased incidence of all EBV-positive lymphomas before the introduction of highly active antiretroviral therapy (HAART) [27]. Between 1996 and 2006, a U.S. registry-based cohort study of 83,282 people with AIDS reported that among cancer-related deaths, 36% of HIV-infected individuals had non-Hodgkin’s lymphoma as the cause, often associated with EBV co-infection [28]. One study on HIV–EBV co-infection, involved reconstituting nonobese diabetic-severe combined immunodeficiency (NOD-*scid*) null mice with human CD34^+^ hematopoietic stem cells, infecting the mice with EBV, and subsequently exposing them to HIV [29]. This exposure led to a transient increase in CD8^+^ T cells observed two weeks after HIV exposure. By the sixth week, the level of EBV-DNA in the plasma was 15 times higher in the subgroup of mice co-infected with HIV compared to the EBV-only group. All co-infected mice developed macroscopic lymphomas with tumors in the spleen, liver, kidneys, gastrointestinal tract, and salivary glands. In contrast, tumors in the EBV-only group were confined to the spleen and liver. This study is consistent with previous work that found a synergistic effect of HIV and EBV in promoting lymphoma development [30].

HIV co-infection suppresses immune responses against EBV, contributing to developing EBV-associated lymphomas [31]. However, the impact of antiretroviral therapy (ART) on EBV-associated lymphomas in HIV-infected individuals remains unclear. While most cases of Hodgkin’s lymphoma in people with AIDS are linked to EBV [32], the incidence of these lymphomas has not declined despite ART usage [33,34,35]. It has even been reported that non-nucleoside reverse transcriptase inhibitors (NNRTIs), a key component of ART regimens, are associated with an increased risk of Hodgkin’s lymphoma [36]. It will be important to determine if NNRTIs do contribute causally to Hodgkin’s lymphoma and, in particular, to those that are EBV-positive.

These results collectively indicate that HIV co-infection enhances EBV-associated lymphomagenesis. Additionally, they illustrate a need to understand any effects of ART on EBV-associated lymphomas in HIV-co-infected patients.

#### 1.3.3. EBV and KSHV Co-Infection of B Cells

Kaposi sarcoma-associated herpesvirus (KSHV), also known as Human Herpesvirus 8 (HHV-8), causes multiple cancers, particularly in sub-Saharan Africa [37]. As with other herpesviruses, KSHV can remain latent in human B cells and endothelial cells, expressing few viral genes and tethering its circular genome to host chromosomes [38,39]. But the conditions that support KSHV infection, particularly in B cells, remain unclear. B cells can only be infected inefficiently in vitro and have not been found to be transformed by the virus [40,41,42,43]. However, more than 80% of PELs worldwide are infected by both EBV and KSHV, with both viral genomes being in the same malignant B cells [44,45]. The infection of B cells by EBV alone in vitro induces their proliferation and can lead to their long-term transformation [46,47,48], outcomes that may explain its contribution to PELs.

The infection of NOD/LtSz-scid IL2Rgamma (NSG) null mice reconstituted with CD34^+^ human hematopoietic progenitor cells with both KSHV and EBV leads to an increase in lymphomas relative to mice infected in parallel with KSHV alone [49]. This finding supports EBV’s fostering lymphomagenesis with KSHV as an etiological co-factor. Infections of primary B cells in vitro with both KSHV and EBV are consistent with this conclusion, too. While only 1–2% of these cells can be co-infected, both viruses are required to initiate and maintain the proliferation of the infected cells in the absence of added cytokines [50,51].

## 2. Sociological Factors Also Affect Lymphomagenesis

Even though EBV can reside in memory B cells asymptomatically for life in most people, some sociological factors can increase the risk of its lymphomagenesis. For example, gender, ethnicity, and the physiological effects of poverty contribute to the incidence of both Hodgkin’s lymphoma and Burkitt’s lymphoma [52,53]. The incidence of these diseases is higher in males compared to females and more prevalent in economically less-developed regions than in more-developed ones. NK/T cell lymphomas are also more common in males and mutations on the X chromosome possibly contribute to this difference [4]. Additionally, the Asian population appears to be more susceptible to EBV-related malignancies, like nasopharyngeal carcinoma and particular forms of gastric carcinoma in immunocompetent individuals [54,55]. However, the correlation with lymphoma remains unclear.

These sociological factors along with human genetics and co-infections work in concert with EBV’s own oncogenic functions to explain its lymphomagenesis. EBV’s success in evolving to infect most people asymptomatically by its controlled latency explains the rarity of these cancers.

## 3. EBV Drives Its Lymphomagenesis in a Background of Impaired Immune Responses

Both a host’s genetics and being co-infected with disparate pathogens can promote EBV’s lymphomagenesis by limiting a host’s immune responses. EBV drives this tumor formation, altering the infected cells and the host with at least three mechanisms: 1. inducing and maintaining cell proliferation; 2. inhibiting cell death; and 3. inhibiting the immune recognition of infected cells. Here, we describe these mechanisms, noting that the viral genes assigned to them reflect only some of what is known now and will surely prove to be an over-simplification (Table 1).

### 3.1. EBV-Induced Proliferation Is Mediated in Part by EBNA2 and LMP1

Early experiments of infecting B cells with EBV demonstrated that the infected cells proliferated in bulk or as clones while the non-infected cells died [47,48,100]. The induced proliferation has been termed “transformation”. This transformation leads rapidly to an increased expression of most cellular and all viral latent genes [101,102]. Analyses of the viral DNAs in these transformed cells indicated both that the viral DNA was present as extrachromosomal replicons and intact [103,104]. Multiple virally encoded RNAs and proteins were detected in these cells, but which ones were required for EBV’s induced proliferation remained unknown until genetic assays were developed to address this uncertainty. The derivatives of the EBV lacking EBNA2 failed to yield proliferating cells [62,63]. A functionally inducible derivative of EBNA2 was used also to demonstrate its requirement for maintaining proliferation in infected B cells [105].

An important insight into EBNA2’s role in mediating the proliferation of EBV-infected B cells came from Henkel et al. [69] who showed that EBNA2 binds to the cellular protein RBJ kappa and they together bind transcriptional promoters to drive RNA expression. RBJ kappa is part of the Notch signaling pathway and EBNA2, through binding, it acts through this pathway [65,66]. In this way, EBNA2 activates many genes, including *c-Myc* to foster proliferation [67,68]. A further increase in expression of c-Myc is common in EBV-associated Burkitt lymphoma in which *c-Myc* is usually translocated to one of three immunoglobulin loci [106]. The translocation leads to the constitutive expression of this oncogene and correlates with the absence of expression of EBNA2 [107]. Thus, EBNA2, while it is required for the proliferation of B cells infected in vitro, can be replaced functionally by mutations acquired by the host–cell. One advantage for the evolving tumor cell to lose expression of EBNA2 would be its ability to escape from the immune recognition of this viral antigen.

EBNA2 also induces the expression of the *LMP1* gene of EBV [108]. LMP1 is essential for the transformation of B cells [109]. It acts as a constitutively active mimic of the CD40 receptor by binding TRAF molecules to signal through NFκB [74,75,76]. The conditional expression of a derivative of LMP1 renders the proliferation of B cells conditional, too, indicating that LMP1 is required also to maintain this proliferation [77]. While LMP1 is usually expressed in EBV-positive Hodgkin’s lymphoma, it is usually absent from Burkitt lymphomas, illustrating again that some EBV-associated lymphomas can evolve to become independent of some viral oncogenes [110,111]. It is likely that the absence of EBNA2 and LMP1 from some EBV-positive lymphomas does reflect an evolution from cells that originally expressed these oncogenes. Analyses in vivo of cells in patients with infectious mononucleosis, which occurs in weeks following infection, found these viral oncogenes to be expressed in the cells [112].

EBV genes, in addition to *EBNA2* and *LMP1*, are expressed in some/all lymphomas including minimally *EBNA1*, *EBER*s, and *miRNA*s. These genes are likely to contribute to the proliferation of the infected B cells in disparate, complex ways. For example, EBNA3A and EBNA3C affect cell proliferation by inhibiting the expression of the tumor suppressors, p16INK4A and p14ARF [72]. EBNA2, through Notch, and LMP1, through NFκB, being essential for cell growth underscore EBV’s commandeering of these cellular pathways to transform lymphoid cells.

### 3.2. EBV’s Inhibition of Apoptosis Is Mediated by Multiple Viral Genes Including EBNA3A, EBNA3C, LMP2A, and Its miRNAs

When B cells are induced to proliferate during normal immune responses, their expansion is limited by a balance of pro- and anti-apoptotic members of the Bcl-2 family [113,114]. EBV overrides this cell death signaling with multiple genes, affecting the distinct modes of apoptosis induced by B cell proliferation. This inhibition of apoptosis is an integral step in EBV’s transformation of B cells. Genetic analyses of EBV found both EBNA3A and EBNA3C to be essential for the initial cell transformation [115], while EBNA3A is not required for its maintenance [64]. One essential function these viral genes provide is the inhibition of the expression of the pro-apoptotic gene *Bim* [116]. EBNA3C, in conjunction with EBNA3A, inhibits the expression of *Bim* by binding its promoter and increasing levels of H3K27me3 locally [117] by recruiting the H3K27 methyltransferase EZH2 to it [118]. EBNA3A and EBNA3C together also inhibit the expression of p16INK4A and p14ARF, two tumor suppressors, that regulate cell cycle progression [72]. This inhibition contributes to the growth of the infected B cells and to the inhibition of Bim-mediated apoptosis [71].

LMP2A is not essential for the EBV’s transformation of B cells isolated from the blood, but it contributes significantly to its efficiency [119]. For the subset of primary B cells isolated from nasal adenoids that do not express the B cell receptor, LMP2A is essential to block apoptosis [82,83,84]. B cells in vivo require tonic signaling mediated by the B cell receptor (BCR) to avoid apoptosis. This signaling indicates to the cell that immunoglobulin is expressed functionally at the cell surface [120]. LMP2A provides this function. It has, for example, been engineered to replace a portion of the immunoglobulin heavy chain locus in transgenic mice so that these B cells do not express the BCR [121]. LMP2A supports the development and survival of these cells in vivo, illustrating its capacity to provide the cell with essential tonic signaling. While LMP2A mimics some facets of the BCR, it has additional properties that contribute to the proliferation of EBV-infected cells [122].

LMP2A’s mimicking BCR tonic signaling is important in EBV’s lymphomagenesis. The tumor cells in Hodgkin’s lymphomas, termed Reed–Sternberg cells, often fail to express a BCR. Most of these tumors are infected with EBV and express LMP2A [123], consistent with EBV being essential for this fraction of Hodgkin’s lymphoma by LMP2A’s tonic signaling inhibiting apoptosis.

Many cellular and viral genes have been identified as potential targets of EBV’s miRNAs [124,125]. Genetic analyses have shown that sets of these miRNAs inhibit apoptosis [86,87] or facilitate growth [91]. Several of EBV’s miRNAs have been shown specifically to target Casp3, a gene product pivotal for initiating apoptosis [88,89,90]. EBV also inhibits apoptosis early after infection. It encodes two relatives of *Bcl-2*, a cellular gene central to inhibiting apoptosis. These genes, *BALF1* and *BHRF1*, have not been found generally to be expressed in B cells transformed in vitro or in tumor cell lines. However, the derivatives of EBV, in which both genes are inactivated, fail to transform cells in vitro [85]. These viral inhibitors of apoptosis are expressed transiently in newly infected B cells, are essential for their initial survival, and are therefore required for their rare evolution into lymphomas.

### 3.3. EBV’s Regulation of Its Host’s Immune Response Is Mediated in Part by EBNA1, EBNA2, EBNA3B, and Its miRNAs

#### 3.3.1. EBNA1

EBNA1 can be the only viral protein expressed in some EBV-positive lymphomas, making it a candidate for affecting the host’s recognition of those tumor cells. It limits its HLA class I-restricted presentation by limiting its proteosomal processing through an extensive Gly-Gly-Ala repeat [58,59,60,61]. It is, however, presented as an exogenous peptide to generate recognition by CD4^+^ T cells [126]. This complex control of its own recognition by the host’s immune response presumably reflects its being essential for the synthesis and partitioning of the viral genome in proliferating cells [2,56].

EBNA1 also directly inhibits the recognition of EBV-infected B cells by inhibiting their expression of the NKG2D ligands ULBP1 and ULBP5 [18]. The infected cells thereby escape efficient recognition by natural killer cells.

#### 3.3.2. EBNA2

While EBNA2 can positively regulate cellular genes, it has also been found to inhibit the expression of some of them. Importantly, when conditional derivatives of EBNA2 are expressed in two different Burkitt lymphoma cell lines, multiple HLA class II genes are inhibited [70]. This inhibition reflects EBNA2’s inhibiting expression of the master regulator of HLA class II gene transcription, *CIITA* (*ibid.*). It should decrease the recognition of infected cells both at early times following initial infection and for the lymphoma cells that express it.

#### 3.3.3. EBNA3B

EBNA3B binds RBPJ sites in the cellular genome in part by association with several transcription factors, as do EBNA3A and EBNA3C [127]. Unlike EBNA3A and EBNA3C, though, it is not needed for the transformation of B cells in vitro [128]. It is, therefore, intriguing to know that it inhibits tumor formation in vivo [73]. When a mutant of EBV deficient in EBNA3B was used to infect NOD-*scid* null mice reconstituted with human immune cells, the animals developed more, larger tumors than those infected with wild-type virus. The lack of EBNA3B did not decrease the generation of CD8^+^ T cells but their infiltration into the tumor sites (*ibid*.). Importantly, human lymphomas often carry variants of EBV mutated in EBNA3B (*ibid.*). Thus, EBNA3B appears to inhibit lymphomagenesis; its mutational inactivation would foster oncogenesis.

#### 3.3.4. miRNAs

EBV encodes more than 40 miRNAs, and many genes have been proposed to be regulated by them [124,125]. Several studies have documented explicit roles for multiple viral miRNAs inhibiting the immune recognition of EBV-infected lymphoid cells. For example, nine viral miRNAs have been shown to target cellular mRNAs involved in antigen-processing, peptide transport, and cytokine synthesis required for CD8^+^ T cell recognition [93]. EBV’s miRNAs also inhibit the recognition of the infected cells by CD4^+^ T cells by inhibiting both antigen presentation and the release of IFN-γ and GM-CSF [94]. One EBV miRNA also targets MICB, a ligand for natural killer cells, to inhibit the killing of infected cells [95]. These findings are consistent with studies in which variants of EBV deleted for most or all of its miRNAs were used to infect NOD-*scid* null mice reconstituted with human immune cells. Infected cells lacking viral miRNAs were inhibited in their growth relative to those infected with wild-type EBV [92].

EBV genes, including *EBNA1*, *EBNA2*, and its *miRNA*s, inhibit the expression of cellular genes that mediate recognition of infected cells by different arms of the human immune response. This inhibition both supports infection in healthy people and can promote lymphomagenesis.

#### 3.3.5. EBERs

The EBERs are two non-coding RNAs that were recognized early as being EBV-encoded and expressed at high levels, even up to 10^7^ molecules per cell [129]. This high level of expression has allowed the detection of the EBERs by in situ hybridization, which has become the go-to clinical method for detecting EBV in infected tissues, in part because these RNAs are also uniformly expressed in infected cells [130]. Early genetic experiments showed that EBER2, not EBER1, was necessary for the efficient transformation of B cells [96], making it likely to play some role in EBV’s lymphomagenesis. Meanwhile, deletion of either one of them correlates to changes in gene expression profile in lymphoblastoid cell lines (LCLs), particularly genes in membrane signaling, apoptosis regulation, and antiviral interferon responses [131], suggesting that the EBERs can keep infected cells from interferon alpha-induced apoptosis [97,98]. EBER2 has subsequently been found to bind the terminal repeats of the viral genome, to recruit PAX5 there, and in so doing, inhibit the expression of LMP2A [132]. EBER2 has also been found to bind the mRNA encoding the cellular deubiquitinase, UCHL1, to increase its expression and enhance the growth of infected cells [99]. EBER2 has thus been found to encode multiple functions, the sum of which contributes to transformation in vitro and, thus, to lymphomagenesis.

### 3.4. EBV Appears to Have the Potential to Be Profoundly Oncogenic; Why Are Its Lymphomas So Rare?

Although the majority of adult humans carry EBV, most never develop EBV-associated malignancies [133]. Why? EBV has evolved to be a “pathobiont”, a virus native to us that only under certain conditions becomes pathogenic and induces disease. It shares this trait with other herpesviruses which have evolved to co-habit successfully with mammalian hosts, an extreme example being Human Herpesvirus 6 (HHV-6). The most common means for congenital infection by HHV-6 is for a newborn to inherit an integrated copy [134]. Multiple features of EBV and its human host foster the virus, being both ubiquitous and non-pathogenic.

## 4. Latency: Trading Pathogenicity for Longevity

EBV can infect a human host, replicate itself and infect many cells, but can progress to reside in some cells quiescently or latently where it expresses only genes that escape immune recognition. Latency allows it to sacrifice continued replication and disease-causing potential for a long-term presence within the host [135]. EBV exhibits four types of latency (Type III, II, I, and 0), following a pattern of successive downregulations of gene expression (Figure 1). In one model, upon infection of naïve B cells, EBV is proposed to express all latent genes, termed Latency III, and progressively restricts protein expression along the path of B cell differentiation, ultimately reaching Latency 0, where no viral proteins are detectable. In this state, the virus persists in recirculating memory B cells, in which it can rarely enter the lytic phase to support its replication [136]. This model of progression, known as the “germinal center model of EBV persistence”, was proposed by Thorley-Lawson [137]. Corresponding to the model, the risk of lymphoma increases when latent B cells are activated and begin expressing viral proteins, as seen in Latency I, II, or III, which can drive cellular proliferation and potentially lead to malignancy. Such activation may result from immunosuppression, chronic immune activation, or mutations. For example, cells in Latency III can be found in healthy carriers but may lead to lymphoma in individuals with compromised immune systems, for example, in patients with post-transplant lymphoproliferative disorders (PTLD) [138]. The most quiescent form, Latency 0, occurs in EBV-positive resting memory B cells in asymptomatic healthy individuals, where no viral proteins are detectable [139]. An alternative to this linear differentiation model [140] posits that EBV can reach Latency 0 without passing through Latency III. This alternate route allows EBV to persist in the memory B cell pool without potentially increasing the risk of lymphomagenesis, optimizing infection without the development of lymphoma.

Whatever the route EBV follows to arrive at Latency 0, it relies on the host’s immune response to eliminate most infected B cells that express detectable viral antigens. Some cells in Latency 0 do likely reactivate, that is, they reenter the productive cycle to replicate viral progeny, so that immunocompetent hosts will be re-stimulated to recognize EBV-infected cells. Approximately 7% of all CD3^+^ T cells in adults target these virally infected cells [141]. The immune system therefore plays a pivotal role in eliminating those EBV-infected cells that express the viral genes which drive proliferation. It is only these cells that can progress to lymphomas.

Current treatments for EBV-associated lymphomas do not usually target EBV. The only viral protein uniformly expressed in these cancers is EBNA1, for which some inhibitors are now being tested [142]. T cells can be educated in vitro and used successfully to treat these lymphomas, but this approach is not yet widely used [143]. Vaccines against EBV are also being developed and are now in clinical trials [144,145,146]. A major therapeutic goal for EBV research is to develop safe inhibitors of its oncogenes, to expand the use of cell-based therapies, and to develop effective vaccines that block its infection. All of these goals are likely to spurred on by the recent appreciation of EBV’s being a risk factor for the development of multiple sclerosis, a chronic disease more common in developed countries than are EBV’s lymphomas [147].

## Figures and Tables

**Figure 1 viruses-16-01744-f001:**
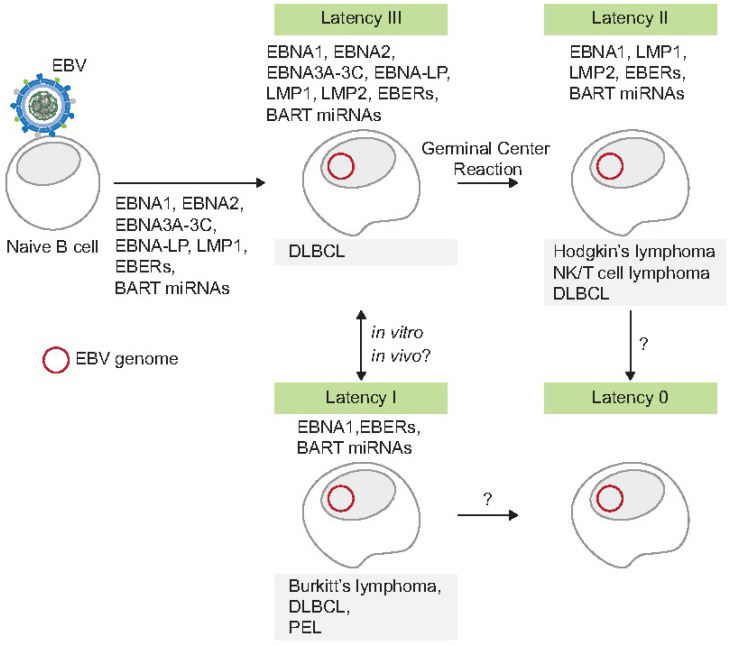
Depicted are possible relationships between cells expressing different sets of EBV genes and the types of lymphomas found to express these subsets. Many of the assays for EBV gene expression also come from analyzing cells grown in vitro so that the subsets of genes expressed in vivo are not certain. The arrows represent possible progressions of cells that express one subset of viral genes to ones that express another subset. How each of these cell types evolves to be a *bona fide* lymphoma must be complex and uncertain. Upon the infection of naïve B cells, EBV expresses all latent genes (Latency III) which appear eventually to be restricted in expression along the path of B cell differentiation, which includes the germinal center reaction. Ultimately, EBV persists in recirculating memory B cells and expresses no detected viral proteins (Latency 0). Latent EBV may be activated by immunosuppression, chronic immune activation, or mutations, and can begin expressing viral proteins (Latency II and III), leading to B cells proliferating and potentially to lymphomagenesis.

**Table 1 viruses-16-01744-t001:** EBV genes and functions implicated in lymphomagenesis.

Gene	Role in Lymphomagenesis	Where Expressed ^#^	Phase
*EBNA1*	- Required for the synthesis and partitioning of the viral genome [2,56,57] - Limits its own HLA class I presentation [58,59,60,61] - Inhibits recognition of infected cells by natural killer cells [18]	HL, BL, DLBCL, NK/T cell lymphoma	Latency I Latency II Latency III
*EBNA2*	- Mediates B cell proliferation through the Notch signaling pathway [62,63,64,65,66,67,68,69] - Inhibits HLA class II presentation [70]		Latency III
*EBNA3A*/*EBNA3C*	- Inhibits apoptosis and foster B cell proliferation by repressing tumor suppressor genes [71,72]	PEL, DLBCL	Latency III
*EBNA3B*	- Inhibits tumor formation [73]	PEL, DLBCL	Latency III
*LMP1*	- Mimics CD40 signaling to promote B cell proliferation through the NFκB pathway [74,75,76,77]	HL, PEL, DLBCL, NK/T-cell lymphoma	Latency II Latency III
*LMP2A*	- Substitutes for B cell receptor signaling and thereby blocks apoptosis [78,79,80,81,82,83,84]	HL, PEL, DLBCL, NK/T-cell lymphoma	Latency II Latency III
*BALF1* and *BHRF1*	- Viral homologs of Bcl-2 that inhibit apoptosis during initial infection of B cells [85]		Early in initial infection
EBV *miRNA*s	- Inhibit apoptosis [86,87,88,89,90] - Facilitate growth of tumor cells [91,92] - Inhibit CD8^+^ T cell recognition [93] - Inhibit CD4^+^ T cell recognition [94] - Inhibit NK cells [95]	All infected cell types	Latency I Latency II Latency III
*EBER*s	- Involved in B cell transformation [96] - Protect infected cells from interferon alpha-induced apoptosis [97,98] - Foster the growth of infected cells [99]	All infected cell types	Latency 0 Latency I Latency II Latency III

^#^ HL, Hodgkin’s lymphoma; BL, Burkitt lymphoma; DLBCL, Diffuse Large B Cell Lymphoma; PEL, Primary Effusion Lymphoma; EBNAs, Epstein–Barr nuclear antigens; LMP, latent membrane protein; miRNAs, microRNAs.

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
