# Peer review of "How Epstein Barr Virus Causes Lymphomas"

_viruses, 2024, doi:10.3390/v16111744_

Round 1
Reviewer 1 Report
Comments and Suggestions for Authors
In this review, the authors summarize the current insights into the mechanism by which the Epstein-Barr Virus (EBV) causes lymphoma. Overall, this manuscript is well-written. Here, I suggest minor modifications to help the readers better understand.
My specific comments are as follows:
1. Chapter 3: In Section 3.1–3.2, the authors describe the roles of EBV genes (EBNA2, LMP1, EBNA3A, and EBNA3B) in cell proliferation and apoptosis suppression in a sentence; however, the function of representative EBV-encoded oncogenes is provided individually in Section 3.3. I recommend that the authors maintain consistency throughout the chapter.
2. I recommend that the authors include EBV-encoded small RNAs (EBER) as a viral gene that contributes to generation of EBV-associated lymphoma, along with appropriate references.
Author Response
Common 1:
Chapter 3: In Section 3.1–3.2, the authors describe the roles of EBV genes (EBNA2, LMP1, EBNA3A, and EBNA3B) in cell proliferation and apoptosis suppression in a sentence; however, the function of representative EBV-encoded oncogenes is provided individually in Section 3.3. I recommend that the authors maintain consistency throughout the chapter.
Ans: We have structured the review with sections 3.1-3.2 being a general introduction and discussed their functions in detail in section 3.3, and do not choose to change it.
Common 2:
I recommend that the authors include EBV-encoded small RNAs (EBER) as a viral gene that contributes to generation of EBV-associated lymphoma, along with appropriate references.
Ans: Thanks for the reviewer’s suggestions. In lines 370-386 (Section 3.3.5 EBERs), Table 1, and the References section (references 96-99; 130-133), the contributions of EBERs to EBV-associated lymphoma and relevant references are included.
Reviewer 2 Report
Comments and Suggestions for Authors
General comments:
This comprehensive review by Chiu Y-F et al summarizes current knowledge on EBV-associated lymphomas, with emphasis on the mechanistic aspects of EBV infection postulated to drive lymphomagenesis. Five lymphomas are discussed, namely Burkitt’s lymphoma, Hodgkin’s lymphoma, NK/T cell lymphoma, diffuse large B-cell lymphoma and primary effusion lymphoma, each of which is associated with one of three latency states. Viral, cellular, and immunological factors involved in the transformation process are also discussed in depth. As a review this paper is current, clearly written and well-referenced.
Specific comments:
1. To help center this paper on EBV-associated lymphomas and factors involved in lymphomagenesis, a definition of lymphoma should be included. Otherwise, lymphoma might be conflated with lymphoproliferative disease post-transplant which is not a lymphoma and has therefore not been discussed in any depth in this paper. Basically, a review of this nature begs for a definition a priori.
2. On Page 4, line 177, the authors discuss male versus female, as well as genetic susceptibility of EBV malignancies in Asians and refer the reader to reference 54. This reference deals with nasopharyngeal carcinoma (NPC), an epithelial cell cancer also driven by EBV infection. This reference may not be pertinent seeing that NPC is mechanistically distinct from EBV-associated lymphomas. Should the authors choose to leave this example in their paper they should at least clearly mention this point of distinction.
3. The nomenclature “MHC” and “HLA” seem to be used interchangeably in this paper. See Table 1 (lines 199 and 206), as well as line 322 in text. To avoid confusion, I suggest that the authors use “HLA” throughout, seeing that they are referring specifically to the human MHC loci.
4. The upper left side of Figure 1 depicts EBV infection of the naïve B cell, listing the genes expressed in early infection. LMP1 expression is not mentioned here. However, there is published literature from at least 2 independent laboratories showing LMP1 production as early as 36 hours following acute infection of primary B cells in vitro (see Alfieri C et al, Virology 1991, 181:595-608). In addition, Allday MJ et al (J gen Virol 1989, 70:1755-64) reported LMP1 expression at 48 hours following in vitro acute infection of primary B cells.
5. The authors conclude that, for EBV-associated lymphomas, the EBV-infected cell is the source of lymphomas. This seems unsurprising as experts on EBV-associated lymphomas would generally agree with this statement. What is missing in terms of a conclusion is a comment on future perspectives within the research field of EBV-associated lymphomas.
6. In terms of form, this review is well-written and very readable. There are, however, a few sentences that require reformulation. For example, on page 1 lines 32-33: “Occasionally EBV DNA is likely to be integrated into a lymphoma cell. Sequencing of it in 128 samples…”. Assuming that I understand this statement correctly, then it should be rewritten as: “Occasionally the EBV genome becomes integrated into the host cell chromosome. Sequencing of integrated EBV DNA in 128 samples…”.
On page 4, lines 158-159 the authors write: “Infection of B cells by EBV alone in vitro induces their proliferation to transform them [45-47] illustrating a likely contribution it makes to PELs.” To improve clarity, I would alter this sentence as follows: “In vitro EBV infection of human primary B-cells induces their activation and long-term transformation, which may explain EBV's contribution to PELs.”
Page 6, line 300: I suggest replacing “it encodes additional properties…” by “it has additional properties…”.
Page 8, line 359: Replace “all of it miRNAs…” by “all of its miRNAs…”.
Author Response
Specific Common 1:
To help center this paper on EBV-associated lymphomas and factors involved in lymphomagenesis, a definition of lymphoma should be included. Otherwise, lymphoma might be conflated with lymphoproliferative disease post-transplant which is not a lymphoma and has therefore not been discussed in any depth in this paper. Basically, a review of this nature begs for a definition a priori.
Ans: Thanks for the reviewer’s suggestions. We have additional descriptions in lines 22-23 to avoid confusion.
Specific Common 2:
On Page 4, line 177, the authors discuss male versus female, as well as genetic susceptibility of EBV malignancies in Asians and refer the reader to reference 54. This reference deals with nasopharyngeal carcinoma (NPC), an epithelial cell cancer also driven by EBV infection. This reference may not be pertinent seeing that NPC is mechanistically distinct from EBV-associated lymphomas. Should the authors choose to leave this example in their paper they should at least clearly mention this point of distinction.
Ans: Thanks for the reviewer’s suggestions. We correct the sentences to make the statement clear: “Additionally, the Asian population appears to be more susceptible to EBV-related malignancies, like nasopharyngeal carcinoma and particular forms of gastric carcinoma in immunocompetent individuals [54, 55]. However, the correlation with lymphoma remains unclear.” (Lines 179-181)
Specific Common 3:
The nomenclature “MHC” and “HLA” seem to be used interchangeably in this paper. See Table 1 (lines 199 and 206), as well as line 322 in text. To avoid confusion, I suggest that the authors use “HLA” throughout, seeing that they are referring specifically to the human MHC loci.
Ans: HLA is used throughout the text. (Table 1 and Lines 326, 337, and 339)
Specific Common 4:
The upper left side of Figure 1 depicts EBV infection of the naïve B cell, listing the genes expressed in early infection. LMP1 expression is not mentioned here. However, there is published literature from at least 2 independent laboratories showing LMP1 production as early as 36 hours following acute infection of primary B cells in vitro (see Alfieri C et al, Virology 1991, 181:595-608). In addition, Allday MJ et al (J gen Virol 1989, 70:1755-64) reported LMP1 expression at 48 hours following in vitro acute infection of primary B cells.
Ans: LMP1 is included in Figure 1, indicating its expression in the early stage of EBV infection.
Specific Common 5:
The authors conclude that, for EBV-associated lymphomas, the EBV-infected cell is the source of lymphomas. This seems unsurprising as experts on EBV-associated lymphomas would generally agree with this statement. What is missing in terms of a conclusion is a comment on future perspectives within the research field of EBV-associated lymphomas.
Ans: Thanks for the reviewer’s suggestions. In lines 429-438 and the References section, an additional conclusion and relevant references (references #143-148) are provided.
Specific Common 6:
In terms of form, this review is well-written and very readable. There are, however, a few sentences that require reformulation. For example, on page 1 lines 32-33: “Occasionally EBV DNA is likely to be integrated into a lymphoma cell. Sequencing of it in 128 samples…”. Assuming that I understand this statement correctly, then it should be rewritten as: “Occasionally the EBV genome becomes integrated into the host cell chromosome. Sequencing of integrated EBV DNA in 128 samples…”.
Ans: Thanks for the reviewer’s suggestion. The sentence is corrected to “Occasionally EBV DNA becomes integrated into the host genome of a lymphoma cell. Sequencing of the EBV’s genome in 128 samples of NK/T cell lymphomas in France and Japan” (Lines 34-36)
On page 4, lines 158-159 the authors write: “Infection of B cells by EBV alone in vitro induces their proliferation to transform them [45-47] illustrating a likely contribution it makes to PELs.” To improve clarity, I would alter this sentence as follows: “In vitro EBV infection of human primary B-cells induces their activation and long-term transformation, which may explain EBV's contribution to PELs.”
Ans: Thanks for the reviewer’s suggestion. The sentence is corrected to “Infection of B cells by EBV alone in vitroinduces their proliferation and can lead to their long-term transformation [46-48], outcomes that may explain its contribution to PELs.” (Lines 160-162)
Page 6, line 300: I suggest replacing “it encodes additional properties…” by “it has additional properties…”.
Ans: Corrected in line 304.
Page 8, line 359: Replace “all of it miRNAs…” by “all of its miRNAs…”.
Ans: Corrected in line 363.